# Graph Representation Learning for Multi-Task Settings: a Meta-Learning Approach

## Abstract

Graph Neural Networks (GNNs) have become the state-of-the-art method for many applications on graph structured data. GNNs are a framework for *graph representation learning*, where a model learns to generate low dimensional node embeddings that encapsulate structural and feature-related information. GNNs are usually trained in an end-to-end fashion, leading to highly specialized node embeddings. While this approach achieves great results in the single-task setting, generating node embeddings that can be used to perform multiple tasks (with performance comparable to single-task models) is an open problem. We propose a novel representation learning strategy, based on meta-learning, capable of producing *multi-task* node embeddings. Our method avoids the difficulties arising when learning to perform multiple tasks *concurrently* by, instead, learning to quickly (i.e. with a few steps of gradient descent) adapt to multiple tasks *singularly*. We show that the embeddings produced by our method can be used to perform multiple tasks with comparable or higher performance than both single-task and multi-task end-to-end models. Our method is model-agnostic and task-agnostic and can hence be applied to a wide variety of multi-task domains.

## 1    Introduction

Graph Neural Networks (GNNs) are deep learning models that operate on graph structured data, and have become one of the main topics of the deep learning research community. Part of their success is given by great empirical performance on many graph-related tasks. Three tasks in particular, with many practical applications, have received the most attention: graph classification, node classification, and link prediction. GNNs are centered around the concept of *node representation learning*, and typically follow the same architectural pattern with an *encoder-decoder* structure (Hamilton et al., 2017; Chami et al., 2020; Wu et al., 2020). The encoder produces node embeddings (low-dimensional vectors capturing relevant structural and feature-related information about each node), while the decoder uses the embeddings to carry out the desired downstream task. The model is then trained in an end-to-end manner, giving rise to highly specialized node embeddings. While this can lead to state-of-the-art performance, it also affects the generalization and reusability of the embeddings. In fact, taking the encoder from a GNN trained on a given task and using its node embeddings to train a decoder for a different task leads to substantial performance loss, as shown in Figure 1.

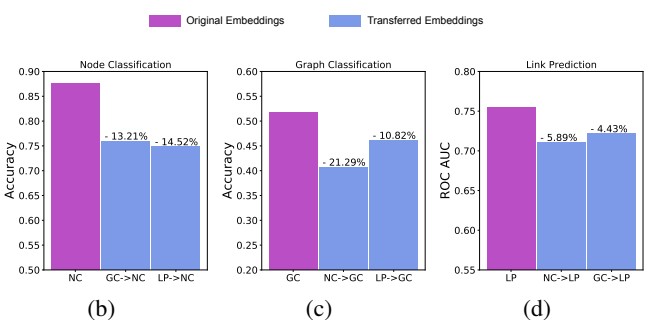

Figure 1: Performance drop when transferring node embeddings between tasks on (a) Node Classification (NC), (b) Graph Classification (GC), and (c) Link Prediction (LP) on the ENZYMES dataset. On the horizontal axis, "*x ->y*" indicates that the embeddings obtained from a model trained on task *x* are used to train a network for task *y*.

The low transferability of node embeddings requires the use of one specialized encoder and one specialized decoder for each considered task. However, many practical machine learning applications operate in resource-constrained environments where being able to share part of the model architecture between tasks is of great importance. Furthermore, the training signal from multiple related tasks can lead to higher generalization. Nevertheless, making sure tasks do not negatively interfere with each other is not trivial (Standley et al., 2020). The problem of learning models that can perform multiple tasks is known as *Multi-Task Learning* (MTL), and is an open area of research, attracting many researchers in the deep learning community (Vandenhende et al., 2020).

MTL on graphs has not received much attention, and no single model capable of performing the three most common graph-related tasks has yet been proposed. In fact, we notice that training a multi-head model with the classical procedure, i.e. by performing multiple tasks concurrently on each graph, and updating the parameters with some form of gradient descent to minimize the sum of the single-task losses, can lead to a performance loss with respect to single-task models. Thus, we propose a novel optimization-based meta-learning (Finn et al., 2017) procedure with a focus on representation learning that can generate node embeddings that generalize across tasks.

Our proposed meta-learning procedure produces *task-generalizing* node embeddings by not aiming at a setting of the parameters that can perform multiple tasks concurrently (like a classical method would do), or to a setting that allows fast multi-task adaptation (like traditional meta-learning), but to a setting that can **easily be adapted to perform each of the tasks singularly**. In fact, our meta-learning procedure aims at a setting of the parameters where a few steps of gradient descent on a given task, can lead to good performance on that task, hence removing the burden of directly learning to solve multiple tasks *concurrently*.

We summarize our contributions as follows:

- We propose a novel method for learning representations that can generalize to multiple tasks. We apply it to the challenging setting of graph MTL, and show that a GNN trained with our method produces higher quality node embeddings with respect to classical end-to-end training procedures. Our method is based on meta-learning and is *model-agnostic* and *task-agnostic*, which makes it easily applicable to a wide range of multi-task domains.

- To the best of our knowledge, we are the first to propose a GNN model generating a *single* set of node embeddings that can be used to perform the three most common graph-related tasks (graph classification, node classification, and link prediction). In particular, our embeddings lead to comparable or higher performance with respect to single-task models even when used as input to a simple linear classifier.

- We show that the episodic training strategy at the base of our proposed meta-learning procedure leads to better node embeddings even for models trained on a single task. This unexpected finding provides interesting directions that we believe can be useful to the whole deep representation learning community.

## 2 RELATED WORK

GNNs, MTL, and meta-learning are very active areas of research. We highlight works that are at the intersections of these subjects, and point the interested reader to comprehensive reviews of each field. To the best of our knowledge there is no work using meta-learning for graph MTL, or proposing a GNN performing graph classification, node classification, and link prediction *concurrently*.

**Graph Neural Networks.** GNNs have a long history (Scarselli et al., 2009), but in the past few years the field has grown exponentially; we refer the reader to Chami et al. (2020); Wu et al. (2020) for a thorough review of the field. The first popular GNN approaches were based on filters in the graph spectral domain (Bronstein et al., 2017), and presented many challenges including high computational complexity. Defferrard et al. (2016) introduced ChebNet, which uses Chebyshev polynomials to produce localized and efficient filters in the graph spectral domain. Graph Convolutional Networks (Kipf & Welling, 2017) then introduced a localized first-order approximation of spectral graph convolutions which was then extended to include attention mechanisms (Veličković et al., 2018). Recently, Xu et al. (2019) provides theoretical proofs for the expressivity of GNNs.

**Multi-Task Learning.** Works at the intersection of MTL and GNNs have mostly focused on multi-head architectures. These models are all composed of a series of GNN layers followed by multiple heads that perform the desired downstream tasks. In this category, Montanari et al. (2019) propose a model for the prediction of physico-chemical properties. Holtz et al. (2019) and Xie et al. (2020) propose multi-task models for concurrently performing node and graph classification. Finally, Avelar et al. (2019) introduce a multi-head GNN for learning multiple graph centrality measures, and Li & Ji (2019) propose a MTL method for the extraction of multiple biomedical relations. The work by (Haonan et al., 2019) introduces a model that can be trained for several tasks singularly, hence, unlike the previously mentioned approaches and our proposed method, it can not perform multiple tasks concurrently. There are also some works that use GNNs as a tool for MTL: Liu et al. (2019b) use GNNs to allow communication between tasks, while Zhang et al. (2018) use GNNs to estimate the test error of a MTL model. We further mention the work by Wang et al. (2020) which considers the task of generating "general" node embeddings, however their method is not based on GNNs, does not consider node attributes (unlike our method), and is not focused on the three most common graph related tasks, which we consider. For an exhaustive review of deep MTL techniques we refer the reader to Vandenhende et al. (2020).

**Meta-Learning.** Meta-Learning consists in *learning to learn*. Many methods have been proposed (see the review by Hospedales et al. (2020)), specially in the area of *few-shot learning*. Garcia & Bruna (2018) frame the few-shot learning problem with a partially observed graphical model and use GNNs as an inference algorithm. Liu et al. (2019a) use GNNs to propagate messages between class prototypes and improve existing few-shot learning methods, while Suo et al. (2020) use GNNs to introduce domain-knowledge in the form of graphs. There are also several works that use meta-learning to train GNNs in few-shot learning scenarios with applications to node classification (Zhou et al., 2019; Yao et al., 2020), edge labelling (Kim et al., 2019), link prediction (Alet et al., 2019; Bose et al., 2019), and graph regression (Nguyen et al., 2020). Finally, other combinations of meta-learning and GNNs involve adversarial attacks on GNN models (Zügner & Günnemann, 2019) and active learning (Madhawa & Murata, 2020).

## 3 PRELIMINARIES

### 3.1 GRAPH NEURAL NETWORKS

Many popular state-of-the-art GNN models follow the *message-passing* paradigm (Gilmer et al., 2017). Let us represent a graph $\mathcal{G} = (\mathbf{A}, \mathbf{X})$ with an adjacency matrix $\mathbf{A} \in \{0, 1\}^{n \times n}$, and a node feature matrix $\mathbf{X} \in \mathbb{R}^{n \times d}$, where the $v$-th row $\mathbf{X}_v$ represents the $d$ dimensional feature vector of node $v$. Let $\mathbf{H}^{(\ell)} \in \mathbb{R}^{n \times d'}$ be the matrix containing the node representations at layer $\ell$. A message passing layer updates the representation of every node $v$ as follows:

$$msg_v^{(\ell)} = \text{AGGREGATE}(\{\mathbf{H}_u^{(\ell)} \, \forall u \in \mathcal{N}_v\})$$
$$\mathbf{H}_v^{(\ell+1)} = \text{UPDATE}(\mathbf{H}_v^{(\ell)}, msg_v^{(\ell)})$$

where $\mathbf{H}^{(0)} = \mathbf{X}$, $\mathcal{N}_v$ is the set of neighbours of node $v$, AGGREGATE is a permutation invariant function, and UPDATE is usually a neural network. After $L$ message-passing layers, the final node embeddings $\mathbf{H}^{(L)}$ are used to perform a given task, and the network is trained end-to-end.

### 3.2 MODEL-AGNOSTIC META-LEARNING AND ANIL

MAML (Model-Agnostic Meta-Learning) is an optimization-based meta-learning strategy proposed by Finn et al. (2017). Let $f_\theta$ be a deep learning model, where $\theta$ represents its parameters. Let $p(\mathcal{E})$ be a distribution over episodes[1], where an episode $\mathcal{E}_i \sim p(\mathcal{E})$ is defined as a tuple containing a *loss function*, a *support set*, and a *target set*: $\mathcal{E}_i = (\mathcal{L}_{\mathcal{E}_i}(\cdot), \mathcal{S}_{\mathcal{E}_i}, \mathcal{T}_{\mathcal{E}_i})$, where support and target sets are simply sets of labelled examples. MAML's goal is to find a value of $\theta$ that can quickly, i.e. in a few steps of gradient descent, be adapted to new episodes. This is done with a nested loop optimization procedure: an *inner loop* adapts the parameters to the support set of an episode by performing some steps of gradient descent, and an *outer loop* updates the initial parameters aiming at a setting that

---

[1]The meta-learning literature usually derives episodes from *tasks* (i.e. tuples containing a dataset and a loss function). We focus on episodes to avoid using the term *task* for both a MTL task, and a meta-learning task.

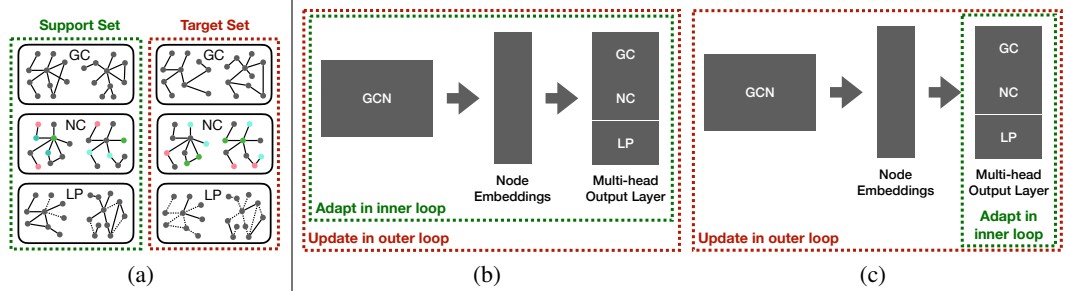

Figure 2: (a) Schematic representation of a *multi-task episode*. For each task, support and target set are designed to be as the training and validation sets for single-task training. (b) Scheme of iSAME: both the backbone GNN and the task-specific output layers are adapted (one at a time) in the inner loop of our meta-learning procedure. (c) Scheme of eSAME: only the task-specific output layers are adapted (one at a time) in the inner loop of our meta-learning procedure.

allows fast adaptation. Formally, by defining $\theta_i'(t)$ as the parameters after $t$ adaptation steps on the support set of episode $\mathcal{E}_i$, we can express the computations in the inner loop as

$$\theta_i'(t) = \theta_i'(t-1) - \alpha \nabla_{\theta_i'(t-1)} \mathcal{L}_{\mathcal{E}_i}(f_{\theta_i'(t-1)}, \mathcal{S}_{\mathcal{E}_i}), \text{ with } \theta_i'(0) = \theta$$

where $\mathcal{L}(f_{\theta_i'(t-1)}, \mathcal{S}_{\mathcal{E}_i})$ indicates the loss over the support set $\mathcal{S}_{\mathcal{E}_i}$ of the model with parameters $\theta_i'(t-1)$, and $\alpha$ is the learning rate. The *meta-objective* that the outer loop tries to minimize is defined as $\mathcal{L}_{meta} = \sum_{\mathcal{E}_i \sim p(\mathcal{E})} \mathcal{L}_{\mathcal{E}_i}(f_{\theta_i'(t)}, \mathcal{T}_{\mathcal{E}_i})$, which leads to the following parameter update[2]

$$\theta = \theta - \beta \nabla_\theta \mathcal{L}_{meta} = \theta - \beta \nabla_\theta \sum_{\mathcal{E}_i \sim p(\mathcal{E})} \mathcal{L}_{\mathcal{E}_i}(f_{\theta_i'(t)}, \mathcal{T}_{\mathcal{E}_i}).$$

Raghu et al. (2020) showed that feature reuse is the dominant factor in MAML: in the adaptation loop, only the last layer(s) in the network are updated, while the first layer(s) remain almost unchanged. The authors then propose ANIL (Almost No Inner Loop) where they split the parameters in two sets: one that is used for adaptation in the inner loop, and one that is only updated in the outer loop. This simplification leads to computational improvements while maintaining performance.

## 4 OUR METHOD

Our novel representation learning technique, based on meta-learning, is built on three insights:

*(i)* **optimization-based meta-learning is implicitly learning robust representations.** The findings by Raghu et al. (2020) suggest that in a model trained with MAML, the first layer(s) learn features that are reusable across episodes, while the last layer(s) are set up for fast adaptation. MAML is then *implicitly* focusing on learning reusable representations that generalize across episodes.

*(ii)* **meta-learning episodes can be designed to encourage generalization.** If we design support and target set to mimic the training and validation sets of a classical training procedure, then the meta-learning procedure is effectively optimizing for generalization.

*(iii)* **meta-learning can learn to quickly adapt to multiple tasks *singularly*, without having to learn to solve multiple tasks *concurrently*.** We can design the meta-learning procedure so that, for each considered task, the inner loop adapts the parameters to a task-specific support set, and tests the adaptation on a task-specific target set. The outer loop then updates the parameters to allow this fast *multiple* **single-task adaptation**. This procedure is effectively searching for a parameter setting that can be easily adapted to obtain good single-task performance, without learning to solve multiple tasks concurrently. This procedure differs from classical training methods (which aim at solving multiple tasks concurrently), and from traditional meta-learning approaches (which aim at parameters that allow fast multi-task adaptation, inheriting the problems of classical methods)[3].

---

[2]We limit ourself to one step of gradient descent for clarity, but any optimization strategy could be used.

[3]We provide a more detailed discussion on the differences with traditional methods in Appendix C.

Based on *(ii)* and *(iii)*, we develop a novel meta-learning procedure where the inner loop adapts to multiple tasks *singularly*, each time with the goal of single-task generalization. Using an encoder-decoder architecture, *(i)* suggests that this procedure leads to an encoder that learns features that are reusable across episodes. Furthermore, in each episode, the learner is adapting to multiple tasks, hence the encoder is learning features that are general across multiple tasks.

***Intuition.*** Training multi-task models is very challenging (Standley et al., 2020), as some losses may dominate over others, or gradients for different tasks may point in opposite directions. Some methods have been proposed to counteract this issues (Kendall et al., 2018; Chen et al., 2018), but they are not always effective and it is not clear how one should choose which method to apply (Vandenhende et al., 2020). We design a meta-learning procedure where the learner does not have to find a configuration of the parameters that *concurrently* performs all tasks, but it has to find a configuration that can **easily be adapted to perform each of the tasks singularly**. By then leveraging the implicit/explicit robust representation learning that happens with MAML and ANIL, we can extract an encoder capable of generating node representations that generalize across tasks.

In the rest of this section, we formally present our novel meta-learning procedure for multi-task graph representation learning. There are three aspects that we need to define: **(1) Episode Design:** how is a an episode composed, **(2) Model Architecture Design:** what is the architecture of our model, **(3) Meta-Training Design:** how, and which, parameters are adapted/updated.

## 4.1 Episode Design

In our case, an episode becomes a *multi-task episode* (Figure 2 (a)). To formally introduce the concept, let us consider the case where the tasks are graph classification (GC), node classification (NC), and link prediction (LP). We define a *multi-task episode* $\mathcal{E}_i^{(m)} \sim p(\mathcal{E}^{(m)})$ as a tuple

$$\mathcal{E}_i^{(m)} = (\mathcal{L}_{\mathcal{E}_i}^{(m)}, \mathcal{S}_{\mathcal{E}_i}^{(m)}, \mathcal{T}_{\mathcal{E}_i}^{(m)})$$

$$\mathcal{L}_{\mathcal{E}_i}^{(m)} = \lambda^{(GC)} \mathcal{L}_{\mathcal{E}_i}^{(GC)} + \lambda^{(NC)} \mathcal{L}_{\mathcal{E}_i}^{(NC)} + \lambda^{(LP)} \mathcal{L}_{\mathcal{E}_i}^{(LP)}$$

$$\mathcal{S}_{\mathcal{E}_i}^{(m)} = \{\mathcal{S}_{\mathcal{E}_i}^{(GC)}, \mathcal{S}_{\mathcal{E}_i}^{(NC)}, \mathcal{S}_{\mathcal{E}_i}^{(LP)}\}, \quad \mathcal{T}_{\mathcal{E}_i}^{(m)} = \{\mathcal{T}_{\mathcal{E}_i}^{(GC)}, \mathcal{T}_{\mathcal{E}_i}^{(NC)}, \mathcal{T}_{\mathcal{E}_i}^{(LP)}\}$$

where $\lambda^{(\cdot)}$ are balancing coefficients. The meta-objective of our method then becomes:

$$\mathcal{L}_{meta}^{(m)} = \sum_{\mathcal{E}_i^{(m)} \sim p(\mathcal{E}^{(m)})} \lambda^{(GC)} \mathcal{L}_{\mathcal{E}_i}^{(GC)} + \lambda^{(NC)} \mathcal{L}_{\mathcal{E}_i}^{(NC)} + \lambda^{(LP)} \mathcal{L}_{\mathcal{E}_i}^{(LP)}.$$

Support and target sets are set up to resemble a training and a validation set. This way the outer loop's objective becomes to *maximize the performance on a validation set, given a training set*, hence pushing towards generalization. In more detail, given a batch of graphs, we divide it in equally sized splits (one per task), and we create support and target sets as follows:

**Graph Classification:** $\mathcal{S}_{\mathcal{E}_i}^{(GC)}$ and $\mathcal{T}_{\mathcal{E}_i}^{(GC)}$ contain labeled graphs, obtained with a random split.

**Node Classification:** $\mathcal{S}_{\mathcal{E}_i}^{(NC)}$ and $\mathcal{T}_{\mathcal{E}_i}^{(NC)}$ are composed of the same graphs, with different labelled nodes. We mimic the common semi-supervised setting (Kipf & Welling, 2017) where feature vectors are available for all nodes, and only a small subset of nodes is labelled.

**Link Prediction:** $\mathcal{S}_{\mathcal{E}_i}^{(LP)}$ and $\mathcal{T}_{\mathcal{E}_i}^{(LP)}$ are composed of the same graphs, with different query edges. In every graph we randomly remove some edges, used as positive examples together with non-removed edges, and randomly sample couples of non-adjacent nodes as negative examples.

The full algorithm for the creation of *multi-task episodes* is provided in Appendix A.

## 4.2 Model Architecture Design

We use an encoder-decoder model with a multi-head architecture. The *backbone* (which represents the encoder) is composed of 3 GCN (Kipf & Welling, 2017) layers with ReLU non-linearities and residual connections (He et al., 2016). The decoder is composed of three *heads*. The node classification head is a single layer neural network with a *Softmax* activation that is shared across nodes

and maps node embeddings to class predictions. In the graph classification head, first a single layer neural network (shared across nodes) performs a linear transformation (followed by a ReLU activation) of the node embeddings. The transformed node embeddings are then averaged and a final single layer neural network with *Softmax* activation outputs the class predictions. The link prediction head is composed of a single layer neural network with a ReLU non-linearity that transforms node embeddings, and another single layer neural network that takes as input the concatenation of two node embeddings and outputs the probability of a link between them.

### 4.3    META-TRAINING DESIGN

We first present our meta-learning training procedure, and successively describe which parameters are adapted/updated in the inner and outer loops.

**Meta-Learning Training Procedure.** To avoid the problems arising from training a model that performs multiple tasks concurrently, we design a meta-learning procedure where the inner loop adaptation and the meta-objective computation involves a *single task* at a time. Only the parameter update performed to minimize the meta-objective involves multiple tasks, but, crucially, it does not aim at a setting of parameters that can solve, or quickly adapt to, multiple tasks *concurrently*, but to a setting that allows multiple **fast single-task adaptation**.

The pseudocode of our procedure is in Algorithm 1. `ADAPT` performs a few steps of gradient descent on a task specific loss function and support set, `TEST` computes the value of the meta-objective component on a task specific loss function and target set, and `UPDATE` optimizes the parameters by minimizing the meta-objective. Notice how the multiple *heads* of the decoder in our model are never used concurrently.

---

**Algorithm 1:** Proposed Meta-Learning Procedure

**Input:** Model $f_\theta$; Episodes $\mathcal{E} = \{\mathcal{E}_1, .., \mathcal{E}_n\}$

`init`$(\theta)$
**for** $\mathcal{E}_i$ *in* $\mathcal{E}$ **do**
    `o_loss` $\leftarrow 0$
    **for** $t$ *in (GC, NC, LP)* **do**
        $\theta'^{(\text{t})} \leftarrow \theta$
        $\theta'^{(\text{t})} \leftarrow \text{ADAPT}(f_\theta, \mathcal{S}_{\mathcal{E}_i}^{(\text{t})}, \mathcal{L}_{\mathcal{E}_i}^{(\text{t})})$
        `o_loss` $\leftarrow$ `o_loss` $+ \text{TEST}(f_{\theta'^{(\text{t})}}, \mathcal{T}_{\mathcal{E}_i}^{(\text{t})}, \mathcal{L}_{\mathcal{E}_i}^{(\text{t})})$
    **end**
    $\theta \leftarrow \text{UPDATE}(\theta, \texttt{o\_loss}, \theta'^{(GC)}, \theta'^{(NC)}, \theta'^{(LP)})$
**end**

---

**Parameter Update in Inner/Outer Loop.** Let us partition the parameters of our model in four sets: $\theta = [\theta_{\text{GCN}}, \theta_{\text{NC}}, \theta_{\text{GC}}, \theta_{\text{LP}}]$ representing the parameters of the backbone ($\theta_{GCN}$), node classification head ($\theta_{NC}$), graph classification head ($\theta_{GC}$), and link prediction head ($\theta_{LP}$). We name our proposed meta-learning strategy SAME (Single-Task Adaptation for Multi-Task Embeddings), and present two variants (Figure 2 (b)-(c)):

*Implicit* **SAME (iSAME):** all the parameters $\theta$ are used for adaptation. This strategy makes use of the *implicit* feature-reuse factor of MAML, leading to parameters $\theta_{\text{GCN}}$ that are general across *multi-task episodes*.

*Explicit* **SAME (eSAME):** only the head parameters $\theta_{\text{NC}}, \theta_{\text{GC}}, \theta_{\text{LP}}$ are used for adaptation. Contrary to the previous, this strategy *explicitly* aims at learning the parameters $\theta_{\text{GCN}}$ to be general across *multi-task episodes* by only updating them in the outer loop.

## 5    EXPERIMENTS

Our goal is to assess the quality of the representations learned by our proposed methods. In more detail, we aim to answer the following questions:

**Q1:** *Do our proposed meta-learning procedures lead to high quality node embeddings in the single-task setting?*

**Q2:** *Do our proposed meta-learning procedures lead to high quality node embeddings in the multi-task setting?*

Table 1: Results for a single-task model trained in a classical supervised manner (Cl), and a **linear** classifier trained on the embeddings produced by our meta-learning strategies (iSAME, eSAME).

| Task | Model | Dataset | | | |
|------|-------|---------|---------|------|------|
| | | ENZYMES | PROTEINS | DHFR | COX2 |
| NC | Cl | $87.5 \pm 1.9$ | $72.3 \pm 4.4$ | $97.3 \pm 0.2$ | $96.4 \pm 0.3$ |
| | iSAME | $87.3 \pm 0.8$ | $81.8 \pm 1.6$ | $96.6 \pm 0.3$ | $96.1 \pm 0.4$ |
| | eSAME | $87.8 \pm 0.7$ | $82.4 \pm 1.6$ | $96.8 \pm 0.2$ | $96.5 \pm 0.6$ |
| GC | Cl | $51.6 \pm 4.2$ | $73.3 \pm 3.6$ | $71.5 \pm 2.3$ | $76.7 \pm 4.7$ |
| | iSAME | $50.8 \pm 2.9$ | $73.5 \pm 1.2$ | $73.2 \pm 3.2$ | $76.3 \pm 4.6$ |
| | eSAME | $52.1 \pm 5.0$ | $72.6 \pm 1.6$ | $71.6 \pm 2.4$ | $75.6 \pm 4.1$ |
| LP | Cl | $75.5 \pm 3.0$ | $85.6 \pm 0.8$ | $98.8 \pm 0.7$ | $98.3 \pm 0.8$ |
| | iSAME | $81.7 \pm 1.7$ | $84.0 \pm 1.1$ | $99.2 \pm 0.4$ | $99.1 \pm 0.5$ |
| | eSAME | $80.1 \pm 3.4$ | $84.1 \pm 0.9$ | $99.2 \pm 0.3$ | $99.2 \pm 0.7$ |

**Q3:** *Do our proposed meta-learning procedures for multiple tasks extract information that is not captured by classically trained multi-task models?*

**Q4:** *Can the node embeddings learned using our proposed meta-learning procedures be used for multiple tasks with comparable or better performance than classical multi-task models?*

Throughout this section we use GC to refer to graph classification, NC for node classification, and LP for link prediction. Unless otherwise stated, accuracy (%) is used for NC and GC, while ROC AUC (%) is used for LP.

**Datasets.** To perform multiple tasks, we consider datasets with graph labels, node attributes, and node labels from the TUDataset library (Morris et al., 2020): ENZYMES (Schomburg et al., 2004), PROTEINS (Dobson & Doig, 2003), DHFR(Sutherland et al., 2003), and COX2 (Sutherland et al., 2003). ENZYMES is a dataset of protein structures belonging to six classes. PROTEINS is a dataset of chemical compounds with two classes (enzyme and non-enzyme). DHFR, and COX2 are datasets of chemical inhibitors which can be active or inactive. In all datasets, each node has a series of attributes containing physical and chemical measurements.

**Experimental Setup.** We perform a 10-fold cross validation, and average results across folds. To ensure a fair comparison, we use the same architecture for all training strategies. We tested loss balancing techniques (e.g. *uncertainty weights* (Kendall et al., 2018), and *gradnorm* (Chen et al., 2018)) but found that they were not effective. In our experiments we notice that the losses do not need to be balanced, and we set $\lambda^{(GC)} = \lambda^{(NC)} = \lambda^{(LP)} = 1$ without performing any tuning. For more information we refer to Appendix B, and we provide source code as supplementary material.

**Q1:** For every task, we train a **linear classifier** on top of the embeddings produced by a model trained using our proposed methods, and compare against a network with the same architecture trained in a classical manner. Results are shown in Table 1. For all three tasks, a **linear** classifier on the embeddings produced by our methods achieves comparable, if not superior, performance to an end-to-end model. In fact, the linear classifier is never outperformed by more than 2%, and it can outperform the classical end-to-end model by up to 12%. These results show that our meta-learning procedures are learning high quality node embeddings.

**Q2:** We train a model with our proposed methods, on all multi-task combinations, and use the embeddings as the input for a **linear classifier**. We compare against models with the same task-specific architecture trained in the classical manner on a single task, and with a fine-tuning baseline. The latter is a model that has been trained on all three tasks, and then fine-tuned on two specific tasks. The idea is that the initial training on all tasks should lead the model towards the extraction of features that it would otherwise not consider (by only seeing 2 tasks), and the fine-tuning process should then allow the model to use these features to target the specific tasks of interest. Results are shown in Table 2 (we omit standard deviation for space limitations). We notice that the embeddings produced by our procedures in a multi-task setting, followed by a **linear** classifier, achieve comparable performance to end-to-end single-task models. In fact, the **linear** classifier is never outperformed by more than 3%, and in 50% of the cases it actually achieves higher performance. We further notice that the fine-tuning baseline severely struggles, and is almost always outperformed by both single-task

Table 2: Results for a single-task model trained in a classical supervised manner, a fine-tuned model (trained on all three tasks, and fine-tuned on the two shown tasks), and a **linear** classifier trained on node embeddings learned with our proposed strategies (iSAME, eSAME) in a multi-task setting.

| Task | | | Dataset | | | | | | | | | | | |
| GC | NC | LP | ENZYMES | | | PROTEINS | | | DHFR | | | COX2 | | |
| | | | GC | NC | LP | GC | NC | LP | GC | NC | LP | GC | NC | LP |
| **Classical End-to-End Training** | | | | | | | | | | | | | | |
| ✓ | | | 51.6 | | | 73.3 | | | 71.5 | | | 76.7 | | |
| | ✓ | | | 87.5 | | | 72.3 | | | 97.3 | | | 96.4 | |
| | | ✓ | | | 75.5 | | | 85.6 | | | 98.8 | | | 98.3 |
| **Fine-Tuning** | | | | | | | | | | | | | | |
| ✓ | ✓ | | 48.3 | 85.3 | | 73.6 | 72.0 | | 66.4 | 92.4 | | 80.0 | 92.3 | |
| ✓ | | ✓ | 49.3 | | 71.6 | 69.6 | | 80.7 | 65.3 | | 58.9 | 80.2 | | 50.9 |
| | ✓ | ✓ | | 87.7 | 73.9 | | 80.4 | 81.5 | | 80.7 | 56.6 | | 87.4 | 52.3 |
| **iSAME (ours)** | | | | | | | | | | | | | | |
| ✓ | ✓ | | 50.1 | 86.1 | | 73.1 | 76.6 | | 71.6 | 94.8 | | 75.2 | 95.4 | |
| ✓ | | ✓ | 50.7 | | 83.1 | 73.4 | | 85.2 | 71.6 | | 99.2 | 77.5 | | 98.9 |
| | ✓ | ✓ | | 86.3 | 83.4 | | 79.4 | 87.7 | | 96.5 | 99.3 | | 95.5 | 99.0 |
| ✓ | ✓ | ✓ | 50.0 | 86.5 | 82.3 | 71.4 | 76.6 | 87.3 | 71.2 | 95.5 | 99.5 | 75.4 | 95.2 | 99.2 |
| **eSAME (ours)** | | | | | | | | | | | | | | |
| ✓ | ✓ | | 51.7 | 86.1 | | 71.5 | 79.2 | | 70.1 | 95.7 | | 75.6 | 95.5 | |
| ✓ | | ✓ | 51.9 | | 80.1 | 71.7 | | 85.4 | 70.1 | | 99.1 | 77.5 | | 98.8 |
| | ✓ | ✓ | | 86.7 | 82.2 | | 80.7 | 86.3 | | 96.6 | 99.4 | | 95.6 | 99.1 |
| ✓ | ✓ | ✓ | 51.5 | 86.3 | 81.1 | 71.3 | 79.6 | 86.8 | 70.2 | 95.3 | 99.5 | 77.7 | 95.7 | 98.8 |

models, and our proposed methods. These results indicate that the episodic meta-learning procedure adopted by SAME is extracting features that are otherwise not accessible with standard training techniques.

**Q3:** We train a multi-task model, and we then train a new simple network (with the same architecture as the heads described in Section 4.2), which we refer to as *classifier*, on the embeddings to perform a task that was not seen during training. We compare the performance of the classifier on the embeddings learned by a model trained in a classical manner, and with our proposed procedure. Intuitively, this tests gives us a way to analyse if the embeddings learned by our proposed approaches contain "more information" than embeddings learned in a classical manner. Results on the ENZYMES dataset are shown in Figure 3, where we notice that embeddings learned by our proposed approaches lead to at least 10% higher performance. We observe an analogous trend on the other datasets, and report all results in Appendix D.

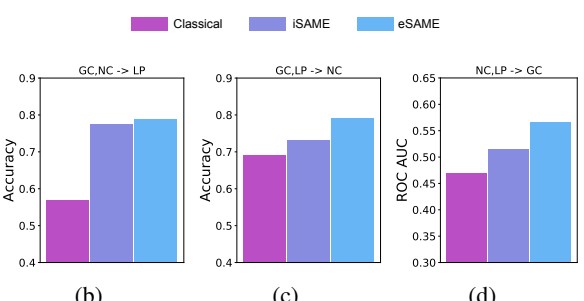

Figure 3: Results for neural network, trained on the embeddings generated by a multi-task model, performing a task that was not seen by the multi-task model. "$x, y \text{->} z$" indicates that $x, y$ are the tasks for training the multi-task model, and $z$ is the new task.

**Q4:** We train the same multi-task model, both in the classical supervised manner, and with our proposed approaches, on all multi-task combinations. For our approaches, we then train a **linear classifier** on top of the node embeddings. We further consider the fine-tuning baseline introduced in **Q2**. We use the multi-task performance ($\Delta_m$) metric (Maninis et al., 2019), defined as the average per-task drop with respect to the single-task baseline: $\Delta_m = \frac{1}{T} \sum_{i=1}^{T} \left( M_{m,i} - M_{b,i} \right) / M_{b,i}$, where $M_{m,i}$ is the value of a task's metric for the multi-task model, and $M_{b,i}$ is the value for the baseline.

Table 3: $\Delta_m$ (%) results for a classical multi-task model (Cl), a fine-tuned model (FT; trained on all three tasks and fine-tuned on two) and a **linear** classifier trained on the node embeddings learned using our meta-learning strategies (iSAME, eSAME) in a multi-task setting.

| Task | | | Model | Dataset | | | |
|---|---|---|---|---|---|---|---|
| GC | NC | LP | | ENZYMES | PROTEINS | DHFR | COX2 |
| | | | Cl | $-0.1 \pm 0.5$ | $4.0 \pm 1.0$ | $-0.3 \pm 0.2$ | $0.5 \pm 0.1$ |
| | | | FT | $-4.5 \pm 1.2$ | $0.1 \pm 0.5$ | $-7.4 \pm 1.4$ | $0.1 \pm 0.4$ |
| ✓ | ✓ | | iSAME | $-2.3 \pm 0.9$ | $2.7 \pm 1.5$ | $-1.2 \pm 0.4$ | $-1.6 \pm 0.2$ |
| | | | eSAME | $-0.8 \pm 0.8$ | $3.2 \pm 1.4$ | $-1.8 \pm 0.3$ | $-1.2 \pm 0.3$ |
| | | | Cl | $-25.3 \pm 3.2$ | $-5.3 \pm 1.2$ | $-28.3 \pm 4.3$ | $-21.4 \pm 3.4$ |
| | | | FT | $-5.1 \pm 1.9$ | $-5.4 \pm 1.5$ | $-24.5 \pm 3.7$ | $-22.6 \pm 3.8$ |
| ✓ | | ✓ | iSAME | $4.1 \pm 0.5$ | $-0.2 \pm 0.9$ | $0.2 \pm 3.2$ | $0.2 \pm 0.5$ |
| | | | eSAME | $3.2 \pm 0.4$ | $-1.2 \pm 1.1$ | $-0.7 \pm 3.4$ | $-0.8 \pm 0.7$ |
| | | | Cl | $7.2 \pm 2.7$ | $6.8 \pm 0.9$ | $-29.1 \pm 7.7$ | $-28.2 \pm 4.5$ |
| | | | FT | $-1.0 \pm 0.3$ | $3.1 \pm 1.2$ | $-28.9 \pm 6.4$ | $-28.3 \pm 4.2$ |
| | ✓ | ✓ | iSAME | $4.4 \pm 1.1$ | $6.1 \pm 1.0$ | $-0.1 \pm 6.2$ | $-0.6 \pm 2.5$ |
| | | | eSAME | $3.9 \pm 1.3$ | $6.1 \pm 1.1$ | $0.1 \pm 6.4$ | $-0.6 \pm 2.6$ |
| | | | Cl | $1.6 \pm 1.3$ | $2.9 \pm 0.3$ | $-18.9 \pm 2.3$ | $-16.9 \pm 3.1$ |
| ✓ | ✓ | ✓ | iSAME | $1.5 \pm 1.0$ | $2.2 \pm 0.2$ | $-0.5 \pm 1.4$ | $-0.9 \pm 1.3$ |
| | | | eSAME | $1.8 \pm 0.9$ | $2.8 \pm 0.2$ | $-1.0 \pm 1.7$ | $-0.4 \pm 1.2$ |

Results are shown in Table 4. We first notice that usually multi-task models achieve lower performance than specialized single-task ones. We then highlight that **linear** classifiers trained on the embeddings produced by our procedures are not only comparable, but in many cases significantly superior to classically trained multi-task models. In fact, a multi-task model trained in a classical manner is highly sensible to the tasks that are being learned (e.g. GC and LP negatively interfere with each other in every dataset), while our methods seem much less sensible: the former has a worst-case average drop in performance of 29%, while our method has a worst-case average drop of less than 3%. Finally, we also notice that the fine-tuning baseline generally performs worst than classically trained models, confirming that transferring knowledge in multi-task settings is not easy, and more advanced techniques, like our proposed method SAME, are needed.

## 6 CONCLUSIONS

In this work we propose a novel representation learning strategy for multi-task settings. Our method overcomes the problems that arise when learning to solve multiple tasks concurrently by optimizing for a parameter setting that can quickly, i.e. with few steps of gradient descent, be adapted for high *single-task* performance on multiple tasks. We apply our method to graph representation learning, and find that our training procedure leads to higher quality node embeddings, both in the multi-task setting, and in the single-task setting. In fact, we show that a linear classifier trained on the embeddings produced by our method has comparable or better performance than classical end-to-end supervised models. Furthermore, we find that the embeddings learned with our proposed procedures lead to higher performance on downstream tasks that were not seen during training. We believe this work can be highly useful to the whole deep representation learning community, as our method is model agnostic and task agnostic, and can therefore be applied on a wide variety of multi-task domains.

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

## A  EPISODE DESIGN ALGORITHM

Algorithm 2 contains the procedure for the creation of the episodes for our meta-learning procedures. The algorithm takes as input a batch of graphs (with graph labels, node labels, and node features) and the loss function balancing weights, and outputs a *multi-task episode*. We assume that each graph has a set of attributes that can be accessed with a *dot-notation* (like in most object-oriented programming languages).

Notice how the episodes are created so that only one task is performed on each graph. This is important as in the inner loop of our meta-learning procedure, the learner adapts and tests the adapted parameters on one task at a time. The outer loop then updates the parameters, optimizing for a representation that leads to fast *single-task adaptation*. This procedure bypasses the problem of learning parameters that *directly* solve multiple tasks, which can be very challenging.

Another important aspect to notice is that the support and target sets are designed as if they were the training and validation splits for training a single-task model with the classical procedure. This way the meta-objective becomes to train a model that can generalize well.

## B  ADDITIONAL EXPERIMENTAL DETAILS

In this section we provide additional information on the implementation of the models used in our experimental section. We implement our models using PyTorch (Paszke et al., 2019), PyTorch Geometric (Fey & Lenssen, 2019) and Torchmeta (Deleu et al., 2019). For all models the number and structure of the layers is as described in Section 4.2 of the paper, where we use 256-dimensional node embeddings at every layer.

At every cross-validation fold the datasets are split into $70\%$ for training, $10\%$ for validation, and $20\%$ for testing. For each model we perform 100 iterations of hyperparameter optimization over the same search space (for shared parameters) using Ax (Bakshy et al., 2018).

We tried some sophisticated methods to balance the contribution of loss functions during multi-task training like GradNorm (Chen et al., 2018) and Uncertainty Weights (Kendall et al., 2018), but we saw that usually they do not positively impact performance. Furthermore, in the few cases where they increase performance, they work for both classically trained models, and for models trained with our proposed procedures. We then set the balancing weights to $\lambda^{(GC)} = \lambda^{(NC)} = \lambda^{(LP)} = 1$ to provide better comparisons between the training strategies.

---

**Algorithm 2:** Episode Design Algorithm

---

**Input** : Batch of $n$ randomly sampled graphs $\mathcal{B} = \{\mathcal{G}_1, .., \mathcal{G}_n\}$
            Loss weights $\lambda^{(GC)}, \lambda^{(NC)}, \lambda^{(LP)} \in [0, 1]$
**Output:** Episode $\mathcal{E}_i = (\mathcal{L}_{\mathcal{E}_i}^{(m)}, \mathcal{S}_{\mathcal{E}_i}^{(m)}, \mathcal{T}_{\mathcal{E}_i}^{(m)})$

$\mathcal{B}^{(GC)}, \mathcal{B}^{(NC)}, \mathcal{B}^{(LP)} \leftarrow$ equally divide the graphs in $\mathcal{B}$ in three sets

```
/* Graph Classification                                                    */
```
$\mathcal{S}_{\mathcal{E}_i}^{\text{(GC)}}, \mathcal{T}_{\mathcal{E}_i}^{\text{(GC)}} \leftarrow$ randomly divide $\mathcal{B}^{(GC)}$ with a 60/40 split

```
/* Node Classification                                                     */
```
**for** $\mathcal{G}_i$ *in* $\mathcal{B}^{(NC)}$ **do**
     `num_labelled_nodes` $\leftarrow \mathcal{G}_i.$`num_nodes` $\times 0.3$
     $\mathcal{N} \leftarrow$ divide nodes per class, then iteratively randomly sample one node per class without
       replacement and add it to $\mathcal{N}$ until $|\mathcal{N}| = $ `num_labelled_nodes`
     $\mathcal{G}_i' \leftarrow$ `copy`$(\mathcal{G}_i)$
     $\mathcal{G}_i.$`labelled_nodes` $\leftarrow \mathcal{N}$;  $\mathcal{G}_i'.$`labelled_nodes` $\leftarrow \mathcal{G}_i.$`nodes` $\setminus \mathcal{N}$
     $\mathcal{S}_{\mathcal{E}_i}^{(NC)}.$`add`$(\mathcal{G}_i)$;  $\mathcal{T}_{\mathcal{E}_i}^{(NC)}.$`add`$(\mathcal{G}_i')$
**end**

```
/* Link Prediction                                                         */
```
**for** $\mathcal{G}_i$ *in* $\mathcal{B}^{(LP)}$ **do**
     $E_i^{(N)} \leftarrow$ randomly pick negative samples (edges that are not in the graph; possibly in the
       same number as the number of edges in the graph)
     $E_i^{1,(N)}, E_i^{2,(N)} \leftarrow$ divide $E_i^{(N)}$ with an 80/20 split
     $E_i^{(P)} \leftarrow$ randomly remove 20% of the edges in $\mathcal{G}_i$
     $\mathcal{G}_i'^{(1)} \leftarrow \mathcal{G}_i$ removed of $E_i^{(P)}$
     $\mathcal{G}_i'^{(2)} \leftarrow$ `copy`$(\mathcal{G}_i'^{(1)})$
     $\mathcal{G}_i'^{(1)}.$`positive_edges` $\leftarrow \mathcal{G}_i'^{(1)}.$`edges`;  $\mathcal{G}_i'^{(2)}.$`positive_edges` $\leftarrow E_i^{(P)}$
     $\mathcal{G}_i'^{(1)}.$`negative_edges` $\leftarrow E_i^{1,(N)}$;  $\mathcal{G}_i'^{(2)}.$`negative_edges` $\leftarrow E_i^{2,(N)}$
     $\mathcal{S}_{\mathcal{E}_i}^{(LP)}.$`add`$(\mathcal{G}_i'^{(1)})$;  $\mathcal{T}_{\mathcal{E}_i}^{(LP)}.$`add`$(\mathcal{G}_i'^{(2)})$
**end**

$\mathcal{S}_{\mathcal{E}_i}^{(m)} \leftarrow \{\mathcal{S}_{\mathcal{E}_i}^{\text{(GC)}}, \mathcal{S}_{\mathcal{E}_i}^{\text{(NC)}}, \mathcal{S}_{\mathcal{E}_i}^{\text{(LP)}}\}$
$\mathcal{T}_{\mathcal{E}_i}^{(m)} \leftarrow \{\mathcal{T}_{\mathcal{E}_i}^{\text{(GC)}}, \mathcal{T}_{\mathcal{E}_i}^{\text{(NC)}}, \mathcal{T}_{\mathcal{E}_i}^{\text{(LP)}}\}$
$\mathcal{L}_{\mathcal{T}_i}^{\text{(GC)}} \leftarrow$ `Cross-Entropy`$(\cdot)$;  $\mathcal{L}_{\mathcal{T}_i}^{\text{(NC)}} \leftarrow$ `Cross-Entropy`$(\cdot)$
$\mathcal{L}_{\mathcal{T}_i}^{\text{(LP)}} \leftarrow$ `Binary Cross-Entropy`$(\cdot)$
$\mathcal{L}_{\mathcal{E}_i}^{(m)} = \lambda^{(GC)}\mathcal{L}_{\mathcal{T}_i}^{\text{(GC)}} + \lambda^{(NC)}\mathcal{L}_{\mathcal{T}_i}^{\text{(NC)}} + \lambda^{(LP)}\mathcal{L}_{\mathcal{T}_i}^{\text{(LP)}}$
**Return** $\mathcal{E} = (\mathcal{L}_{\mathcal{E}_i}^{(m)}, \mathcal{S}_{\mathcal{E}_i}^{(m)}, \mathcal{T}_{\mathcal{E}_i}^{(m)})$

---

**Linear Model.** The linear model trained on the embeddings produced by our proposed method is a standard linear SVM. In particular we use the implementation available in Scikit-learn (Pedregosa et al., 2011) with default hyperparameters. For graph classification, we take the mean of the node embeddings as input. For link prediction we take the concatenation of the embeddings of two nodes. For node classification we keep the embeddings unaltered.

**Deep Learning Baselines.** We train the single task models for 1000 epochs, and the multi-task models for 5000 epochs, with early stopping on the validation set (for multi-task models we use the sum of the task validation losses or accuracies as metrics for early-stopping). Optimization is done using Adam (Kingma & Ba, 2015). For node classification and link prediction we found that normalizing the node embeddings to unit norm in between GCN layers helps performance.

**Our Meta-Learning Procedure.** We train the single task models for 5000 epochs, and the multi-task models for 15000 epochs, with early stopping on the validation set (for multi-task models we use the sum of the task validation losses or accuracies as metrics for early-stopping). Early stopping is very important in this case as it is the only way to check if the meta-learned model is overfitting the training data. The inner loop adaptation consists of 1 step of gradient descent. Optimization in the outer loop is done using Adam (Kingma & Ba, 2015). We found that normalizing the node embeddings to unit norm in between GCN layers helps performance.

## C  COMPARISON WITH TRADITIONAL TRAINING APPORACHES

Our proposed meta-learning approach is significantly different from the classical training strategy (Algorithm 3), and the traditional meta-learning approaches (Algorithm 4).

The classical training approach for multi-task models takes as input a *batch* of graphs, which is simply a set of graphs, where on each graph the model has to execute *all* the tasks. Based on the cumulative loss on all tasks

$$\mathcal{L} = \lambda^{(GC)}\mathcal{L}^{(\text{GC})} + \lambda^{(NC)}\mathcal{L}^{(\text{NC})} + \lambda^{(LP)}\mathcal{L}^{(\text{LP})}$$

for all the graphs in the batch, the parameters are updated with some form of gradient descent, and the procedure is repeated for each batch.

The traditional meta-learning approach takes as input an episode, like our approach, but for every graph in the episode *all* the tasks are performed. The support set and target set are *single* sets of graphs, where every task can be performed on all graphs. The support set is used to obtain the adapted parameters $\theta'$, which have the goal of *concurrently* solving all tasks on all graphs in the target set. The loss functions, both for the inner loop and for the outer loop, are the same as the one used by the classical training approach. The outer loop then updates the parameters aiming at a setting that can easily, i.e. with a few steps of gradient descent, be adapted to perform multiple tasks *concurrently* given a support set.

| **Algorithm 3:** Classical Training | **Algorithm 4:** Traditional Meta-Learning |
|---|---|
| **Input:** Model $f_\theta$; Batches $\mathcal{B} = \{\mathcal{B}_1, .., \mathcal{B}_n\}$
`init`$(\theta)$
**for** $\mathcal{B}_i$ *in* $\mathcal{B}$ **do**
$\quad$`loss` $\leftarrow$ concurrently perform all tasks
$\quad\quad$ on all graphs in $\mathcal{B}_i$
$\quad \theta \leftarrow$ `UPDATE`$(\theta, \text{loss})$
**end** | **Input:** Model $f_\theta$; Episodes $\mathcal{E} = \{\mathcal{E}_1, .., \mathcal{E}_n\}$
`init`$(\theta)$
**for** $\mathcal{E}_i$ *in* $\mathcal{E}$ **do**
$\quad$`i_loss` $\leftarrow$ concurrently perform all
$\quad\quad$ tasks on all support set graphs
$\quad \theta' \leftarrow$ `ADAPT`$(\theta, \text{i\_loss})$
$\quad$`o_loss` $\leftarrow$ concurrently perform all
$\quad\quad$ tasks on all target set graphs using
$\quad\quad$ parameters $\theta'$
$\quad \theta \leftarrow$ `UPDATE`$(\theta, \theta', \text{o\_loss})$
**end** |

Table 4: Results of a neural network trained on the embeddings generated by a multi-task model, to perform a task that was not seen during training by the multi-task model. "$x,y ->z$" indicates that the multi-task model was trained on tasks $x$ and $y$, and the neural network is performing task $z$.

| Task | Model | Dataset | | | |
|---|---|---|---|---|---|
| | | ENZYMES | PROTEINS | DHFR | COX2 |
| | Cl | $56.9 \pm 3.9$ | $54.4 \pm 1.4$ | $61.2 \pm 2.2$ | $59.8 \pm 0.4$ |
| GC,NC ->LP | iSAME | $77.3 \pm 4.5$ | $88.5 \pm 1.8$ | $99.8 \pm 1.8$ | $97.1 \pm 2.0$ |
| | eSAME | $78.9 \pm 2.8$ | $89.1 \pm 1.5$ | $99.7 \pm 2.2$ | $95.8 \pm 3.3$ |
| | Cl | $69.1 \pm 1.2$ | $57.3 \pm 1.6$ | $58.3 \pm 9.3$ | $68.9 \pm 10.7$ |
| GC,LP ->NC | iSAME | $73.3 \pm 2.1$ | $59.2 \pm 2.5$ | $77.6 \pm 1.6$ | $78.1 \pm 4.6$ |
| | eSAME | $79.1 \pm 1.7$ | $64.7 \pm 3.0$ | $76.1 \pm 2.7$ | $76.9 \pm 3.3$ |
| | Cl | $47.1 \pm 2.4$ | $75.3 \pm 1.5$ | $77.5 \pm 3.1$ | $79.9 \pm 3.4$ |
| NC,LP ->GC | iSAME | $48.5 \pm 5.5$ | $76.1 \pm 2.3$ | $76.1 \pm 3.7$ | $79.7 \pm 5.1$ |
| | eSAME | $56.6 \pm 3.1$ | $74.6 \pm 2.7$ | $77.1 \pm 3.6$ | $79.3 \pm 6.2$ |

## D  FULL RESULTS FOR THE GENERALIZATION OF NODE EMBEDDINGS

Table 4 contains results for a neural network, trained on the embeddings generated by a multi-task model, to perform a task that was not seen during the training of the multi-task model. Accuracy (%) is used for node classification (NC) and graph classification (GC); ROC AUC (%) is used for link prediction (LP). The embeddings produced by our meta-learning methods lead to higher performance (up to **35%**), showing that our procedures lead to the extraction of more informative node embeddings with respect to the classical end-to-end training procedure.

