# OpenReview forum: "Graph Representation Learning for Multi-Task Settings: a Meta-Learning Approach"
_ICLR.cc/2021/Conference — Reject_

### Official Review · AnonReviewer2 · 2020-10-28
**Graph Representation Multi-Task Learning.**

**Rating:** 7
**Confidence:** 3

**Review:**

This paper presents a multi-task framework to represent the node embedding for transferred knowledge. The methodology is based on the meta-learning, which is capable of producing multi-task node embedding. This paper is well-motivated and well-written. The experimental results illustrate the effectiveness of the model. I would like to recommend to accept this paper.

Major Concerns:
1. The related work can be strengthened. In the current version, the related work seems to stack all the papers in sequence, which makes it tough to understand the development of this task. I suggest the authors to reformate this subsection to Mind Graph.
2. Can you introduce the dataset more specifically?
3. There shall be more baselines for the experiments.

Minor Concern:
1. Fig.1: the legend shall be outside the box.
2. Sec 3 seems redundant to most related readers.

---

> ### Author Response · Authors · 2020-11-15
> **Answer to Reviewer 2**
>
> We thank the reviewer for the positive and insightful comments and we provide answers to his questions below.
>
> ### Q1:
> The initial version of the Related Work section was forced by space limitations, we now provide a revised version of the paper with an extended Related Work section.
>
> ### Q2:
> We had to limit our presentation of the datasets for space limitations, but the new version of the paper that we uploaded follows the comments of the reviewer and provides more information on the datasets.
>
> ### Q3:
> The new version of our paper adds a fine-tuning baseline, where a multi-task model is first trained on the three considered tasks, and then it is fine-tuned on specific task(s) of interest.  The reasoning behind this is that the initial training on all tasks should lead the model towards the extraction of features that it would otherwise not consider (by only seeing 2 tasks), and the fine-tuning procedure then allows it to use these features to target the specific tasks of interest. Results in Table 2 and Table 3 show that this method is outperformed by SAME and actually performs worse than classically trained models. This shows that, in order to extract general information that can be effectively exploited for multiple tasks, we need new representation learning procedures like SAME, as traditional training methods are not enough.
>
> ### Minor Comments
> We thank again the reviewer for the comments. We fixed the legend in Figure 1 and in Figure 3. We believe Section 3 can be useful to people that are not familiar with Graph Neural Networks, MAML, and ANIL, and acts as a refresher (for the more experienced readers) that increases the readability of the paper.

---

### Official Review · AnonReviewer3 · 2020-10-30
**Direct extension of MAML**

**Rating:** 5
**Confidence:** 4

**Review:**

This paper formulates the learning of three tasks, including graph classification, node classification and link prediction, as a multi-task learning problem and adopts a meta learning approach to learn the three tasks together in the spirit of the Model-Agnostic Meta Learning (MAML) method.

Actually, there have been some works to study the three tasks (i.e., graph classification, node classification and link prediction) as a multi-task learning problem. Authors need to discuss differences with those works and compare with them in experiments.

The proposed meta learning approach seems a direct application of the MAML method. I cannot see much difference with the MAML method.

In the meta-objective, how to set different \lambda’s? This is more important to the performance.

---

> ### Author Response · Authors · 2020-11-15
> **Answer to Reviewer 3**
>
> We thank the reviewer for his insights and we answer below to his questions and comments.
> ### Q1
> We are not aware of works that study the use of Graph Neural Networks for graph classification, node classification, and link prediction concurrently in a multi-task setting. We kindly ask the reviewer if he can provide the references to such works, if we have not mentioned them in this answer. We are in fact aware of some papers that, at a high level, may seem related to ours, and we provide below the motivations as to why they are not. We further add these comments and references in the new version of the paper.
> - Graph Star Net for Generalized Multi-Task Learning, Haonan et al., arXiv 2019: the authors propose a model that can be trained for multiple tasks, however it can be trained only on one task at a time and it can not perform multiple tasks concurrently. It is in fact a single-task model, as, once it is trained, it can only perform one task (the one it was trained for), and it can be trained only on one task at a time.
> - Multi-Task Network Representation Learning, Xie et al., Frontiers in Neuroscience 2020: this paper proposes a multi-head model that can perform graph classification and node classification. Not only we already cover multi-head models in our experiments, but the proposed method is also not capable of performing link prediction (we already referenced this paper in our initial submission).
> - Multi-Task Learning Based Network Embedding Wang et al., Frontiers in Neuroscience 2020: the authors propose a method that does not use Graph Neural Networks, does not consider node attributes, and does not perform the three tasks we perform. Furthermore, it makes use of "handcrafted" features to extract structural information, and in fact the authors do not provide comparisons to end-to-end models like Graph Neural Networks.
>
> ### Q2
> There are several important differences between MAML and SAME. We provide a summary of such differences below:
> - In most existing applications of MAML (like the ones for few-shot learning), episodes are all instantiations of the same task, i.e. in each episode the same loss function is used and the same number of support examples is used (e.g., the k-shot n-way framework). In fact, in previous uses of meta-learning (specially the few-shot learning ones), the difference between episodes is only given by the different labels of the examples involved. In our method, not only we have multiple tasks in each episode, but each task can have a different loss function and a different number of training examples. It is actually possible to view MAML as a special case of SAME, where we only have one task with a fixed structure.
> - MAML and other variations of MAML perform a single adaptation on the support set in the inner loop. In SAME, we perform multiple adaptations (one per task), each time starting from the same initial parameter configuration. Furthermore, in each adaptation, different subsets of parameters are involved, which again is a property that differentiates us from previous approaches, and actually poses SAME as a generalization of MAML.
>
> ### Q3
> We thank the reviewer for the comment. For space motivations we only discussed this aspect in the Appendix, but the new version of the paper has been updated to contain comments about the \lambda parameters. In our experiments we saw that the \lambdas were not very important for SAME. In fact we set them all to 1 and did not optimize them. We did introduce them in our presentation as we wanted to provide a general framework that could be instantiated in many different ways.

---

### Official Review · AnonReviewer1 · 2020-10-31
**interesting  problem**

**Rating:** 6
**Confidence:** 5

**Review:**

The manuscript proposes SAME, a model based on GNN and meta-learning for learning multi-task node embeddings. Unlike multi-task learning setting, SAME aims at learning to quickly adapt to multiple tasks. Two model variants iSAME and eSAME are proposed base on different settings in inner/outer loop of parameter update. Experiments on several datasets demonstrate the good performance of SAME.

Pros
1. The problem is new and interesting. It is the first work to study single set of node embeddings for multi-tasks in graph.
2. The presentation is overall good. The content is clear for me.
3. Introduce a new model for learning multi-task node embeddings through meta-learning way. The model is simple yet interesting for new problem.

Cons/Questions
1. The novelty of this work incremental. Despite the new problem and different task settings, the model framework adopts the similar procedure as general meta-learning procedure. The model is quite simple. I would like to see more discussion about the contribution and novelty of this work as well as the potential future study.

2. This work follows the meta-learning setting. Besides studying different graph learning tasks, it is better to provide content and add experiment for the scenario of few-shot labeled data. If I did not miss it, there is no discussion and experiment about this. I would suggest the authors to add comparison experiments for different tasks where only few-shot supervised data are available.

3. Reference format is not consistent, typos, etc.

---

> ### Author Response · Authors · 2020-11-15
> **Answer to Question 1**
>
> We thank the reviewer for the insightful comments and we provide answers to his comments below. (For character limitations we answer to question 1 below, and to question 2 and 3 in a separate comment).
>
> ### Q1:
> Our work does fit into the meta-learning framework composed of a nested loop optimization, but presents several important aspects that differentiates it from previously proposed methods. We first provide more information on how our work positions itself in the Meta-Learning literature and what are the differences from existing methods, and,  successively, we highlight several interesting directions for future work.
>
> #### Novelty of SAME
> - Meta-learning has been used in many scenarios such as few-shot learning, continual learning, domain adaptation, and autoML. However, the use of meta-learning as a technique for representation learning (as we do) was entirely unexplored before our work, and, further, we do it for the multi-task setting.
> - In meta-learning the task design is a very important factor. We substantially differentiate ourself from most previous works, as we design target and support sets to contain multiple tasks and to resemble training and validation sets.
> - In most existing applications of meta-learning (like the ones for few-shot learning), episodes are all instantiations of the same task, i.e. in each episode the same loss function is used and the same number of support examples is used (e.g., in the k-shot n-way framework). In fact, in previous uses of meta-learning (specially the few-shot learning ones), the difference between episodes is only given by the different labels of the examples involved. In our method, not only we have multiple tasks in each episode, but each task can have a different loss function and a different number of training examples. It is actually possible to view MAML as a special case of SAME, where we only have one task with a fixed structure in both support and target sets.
> - MAML and its variations proposed in previous work perform a single adaptation on the support set in the inner loop. In SAME, we perform multiple adaptations (one per task), each time starting from the same initial parameter configuration. Furthermore, in each adaptation, different subsets of parameters are involved, which again is a property that differentiates us from previous approaches, and actually poses SAME as a generalization of MAML.
> - As pointed out by the reviewer, the GNN model we use is quite simple, and we believe this is a strong positive point for our work. If our method could work only with sophisticated architecture, then it would be of limited use. The fact that it works very well with such a simple model is a promising signal for the future applications of the method proposed in our work.
>
> #### Future Works
> The generality of the SAME framework offers many directions for future work. Some of the most interesting ones are presented below.
> - As written above, we are the first to view meta-learning as a tool for representation learning. This new view of meta-learning allows us to design new ways to extract information from data by properly engineering the meta-learning episodes. We believe this new view can be of great interest to the community.
> - It is possible to explore the use of more sophisticated architectures, and the combinations of other tasks for multi-task learning. This also includes applications outside of the Graph domain.
> - Another area for exploration is in the design of more sophisticated episodes, where the task(s) for adaptation in the inner loop are different from the task(s) in the outer loop. E.g., the inner loop could perform multiple low-level tasks, while the outer loop could perform higher-level tasks. This way the learner has to extract information from the low-level tasks that can be used to better perform higher level tasks. And this could lead to very powerful new ways to learn representations.
> - We follow ANIL and its results by only using the last layers for adaptation in the inner loop (in iSAME). However, it is interesting to explore more sophisticated choices for selecting the parameters used for adaptation. There could in fact be some multi-task combinations, and some sophisticated architectures, where it is actually best to subdivide the parameters in different ways.
> - Finally, we want to highlight the direction of applications to few-shot learning (as suggested by the reviewer), and other areas where meta-learning has proven itself useful (e.g. continual learning).

---

> ### Author Response · Authors · 2020-11-15
> **Answer to Questions 2 & 3**
>
> ### Q2:
> Meta-learning has proven to be useful in many scenarios: few-shot learning, continual learning, domain adaptation, autoML. In this paper we focus on the unexplored representation learning scenario. In fact, we are the first to view meta-learning as a technique for representation learning, and we focus our work on the specific task of graph representation learning. On this aspect, we added another baseline (a fine-tuned model) in our experiments to further confirm that standard training techniques are not enough to properly extract general features in multi-task scenarios. While the few-shot learning setting is definitely an interesting one, we believe it is a promising direction for future work that necessitates a paper of its own to be properly explored.
>
> ### Q3:
> We thank the reviewer for pointing this out. We provide an updated version of our paper with a revised reference format.

---

### Author Response · Authors · 2020-11-15
**Summary of changes made to paper**

We thank the reviewers for their insightful reviews. We did our best to answer all the concerns and we are available for further questions. We have uploaded a new version of the paper implementing the reviewers' comments. We summarizes the changes as follows:
- We reformulated the Related Work section, and we added more references. (Section 2)
- We revised the reference format. (References)
- We provide a brief description of each dataset. (Section 5)
- We added results for a Fine-Tuning baseline. (Section 5)
- In our experimental section we added a brief discussion on the lambda hyperparamters. (Secction 5)

---

### Decision · Program_Chairs · 2021-01-07
**Final Decision**

**Decision:**

Reject

**Comment:**

This paper experimentally observes the negative transfer in Multi-task Graph Representation Learning and proposes to solve the negative transfer with a novel Meta-Learning based training procedure. However, the proposed methods seems not technically sound. There are some concerns about this paper：1. The technique contribution of this paper is limited. The method proposed in this paper is just an application of MAML in Graph Representation Learning with a little variation. 2. This paper only compares SAME with the vanilla MTL method, which adopts the uniform weights. However, the vanilla MTL method commonly performs poorly. The state-of-the-art MTL methods should be taken into comparison, for example MGDA [1]. 3. The traditional Meta-Learning framework introduced in Algorithm 4 is misleading.  4. The experimental analysis of this paper is not sufficient. For example, the paper has not analyzed whether the improvement comes from the meta updating or comes from the singularly training strategy.

[1]. Sener, Ozan, and Vladlen Koltun. "Multi-task learning as multi-objective optimization." NIPS 2018.